# Enhancement of Gel Strength of Itaconic Acid-Based Superabsorbent Polymer Composites Using Oxidized Starch

**DOI:** 10.3390/polym13172859

**Published:** 2021-08-25

**Authors:** Haechan Kim, Jungsoo Kim, Donghyun Kim

**Affiliations:** 1Material & Component Convergence R&D Department, Korea Institute of Industrial Technology (KITECH), Ansan 15588, Korea; coolskawk@kitech.re.kr (H.K.); kimjungsoo11@kitech.re.kr (J.K.); 2Department of Material Chemical Engineering, Hanyang University, Ansan 15588, Korea

**Keywords:** superabsorbent polymer composites, surface-crosslinking, ethylene glycol diglycidyl ether, absorption properties, re-swellability

## Abstract

Herein, core-superabsorbent polymer (CSAP) composites are prepared from oxidized starch (OS) via aqueous solution copolymerization using ammonium persulfate as the initiator, and 1,6-hexanediol diacrylate as the inner-crosslinker. The surface-crosslinking process is performed using various surface-crosslinkers, including bisphenol A diglycidyl ether (BADGE), poly(ethylene glycol) diglycidyl ether (PEGDGE), ethylene glycol diglycidyl ether (EGDGE), and diglycidyl ether (DGE). The structures of the CSAP composites and their surface-crosslinked SAPs (SSAPs) are characterized using Fourier transform infrared (FT-IR) spectroscopy, their absorption properties are measured via centrifuge retention capacity (CRC), absorbency under load (AUL), permeability, and re-swellability tests, and their gel strengths according to surface-crosslinker type and EGDGE content are examined via rheological analysis. The results indicate that an EGDGE content of 0.75 mol provides the optimum surface-crosslinking and SSAP performance, with a CRC of 34.8 g/g, an AUL of 27.2 g/g, and a permeability of 43 s. The surface-crosslinking of the CSAP composites using OS is shown to improve the gel strength, thus enabling the SAP to be used in disposable diapers.

## 1. Introduction

Superabsorbent polymers (SAPs) are hydrogels in which the absorption characteristics are maximized via a light crosslinking density that enables expansion of the three-dimensional network of chains to capture 10 to 1000 times more water than standard adsorbent polymers, depending on the specific absorption medium [1]. The applications of SAPs are mainly in the field of hygiene, such as in water-absorbing materials for disposable diapers and women’s sanitary goods, where they can capture secreted fluids such as urine, blood, etc. [1]. In addition, they are widely applied in the non-hygiene fields such as in water-swellable rubbers, drug-delivery systems [2], soil additives for agriculture [1,3], and sensors of environmental pollutants [1,2,3], etc.

Many researchers have attempted to utilize biomass or readily-available natural polymers such as starch and cellulose in SAPs [4,5,6]. The SAP synthesis route for natural polymers involves the graft copolymerization of vinyl monomers onto the natural polymer chains [7]. However, this approach has limitations in the amount chain extension from macromolecules, so that attention has focused more on the production of SAPs from biomass monomers. In this respect, itaconic acid (IA) is a promising biomass material with high added value. It is rarely observed in nature, but is produced industrially by the fermentation of carbohydrates such as lactic acid and starch, or by the combustion of natural citric acid [8]. The IA molecule contains two carboxyl groups and one vinyl group, the latter forming part of a vinyl acid moiety similar to that of acrylic acid (AA). However, the absorption performance of the IA-based SAPs in hygienic applications is limited by their low gel strength [9,10], and attempts to improve their gel strengths have been conducted using silica [11], hydroxyapatite [12], montmorillonite [13], and bentonite [14].

Oxidized starch (OS) modifies the hydroxyl group of the starch molecule through various catalysts, or its molecular weight is reduced through cleavage of α-1,4 bonds (glycosidic bonds). It has been reported that the gelation properties of starch can be controlled according to the oxidation ratio of starch [15].

Surface-crosslinking is a process for manufacturing core-shell structured SAPs via additional crosslinking reactions on the surface of the primary polymerized core-SAP (CSAP). Thus, the core-shell SAP structure is designed to include an inner CSAP composite with a low crosslinking density and an outer CSAP composite with a high crosslinking density [16], along with enhanced gel strength, high absorption capacity under pressure, and resistance to gel-blocking phenomena [17].

Studies of surface-crosslinker have been reported on diols that induce ester bonds with carboxyl groups of polymer chains, and other examples include epoxy [18] and amine [19].

In the present study, CSAP composites are prepared via the in-situ copolymerization of IA and AA with OS as the filler. We evaluated low molecular weight OS’s applicability as a filler in SAP networks. The resulting IA-based CSAP is then treated with various surface-crosslinkers including bifunctional epoxies to obtain the requisite excellent gel strength for application in disposable diapers. Surface-crosslinkers with epoxy groups are expected to be highly efficient through fast reaction rates and low temperature processes.

## 2. Materials and Methods

### 2.1. Materials

The IA (≥99%, Junsei Chemical, Tokyo, Japan), acrylic acid (AA, 99%, Sigma Aldrich, Saint Louis, MO, USA), OS (99%, Daejung Chemicals, Siheung, Korea), and 50% aqueous sodium hydroxide solution (Samchun Pure Chemicals, Pyeongtaek, Korea) were used as received without further purification. Ammonium persulfate (APS, reagent grade, 98%, Sigma Aldrich) was used as the initiator; 1,6-hexanediol diacrylate (HDODA, 99%, Sigma Aldrich) was used as inner-crosslinker without purification. Methanol (95%, Samchun Pure Chemicals), bisphenol A diglycidyl ether (BADGE, 99%, Sigma Aldrich), poly(ethylene glycol) diglycidyl ether (PEGDGE, M_n_ = 275, Sigma Aldrich), ethylene glycol diglycidyl ether (EGDGE, 98%, Sigma Aldrich), and diglycidyl ether (DGE, 98%, Sigma Aldrich) were used as the surface-crosslinkers without purification.

### 2.2. Preparation of the CSAP Composites

The partially neutralized poly(IA-*co*-AA) was synthesized by aqueous solution polymerization using APS as the initiator [20]. The reactions were performed in a 500-mL four-necked flask equipped with a mechanical stirrer, a reflux condenser, a thermometer, and a needle for the injection of nitrogen gas at 60 °C. For each reaction, IA (0.5 mol) and AA (0.5 mol) were stirred in 90 mL of 50% aqueous NaOH solution at 250 rpm to achieve a 75% degree of neutralization of the carboxyl groups to sodium carboxylate. This was followed by the addition of APS along with HDODA (2.0 mmol%), and selected proportions of OS and HDODA, as listed in Table 1. After 3 h, the obtained gel was dried in a convection oven at 60 °C for 24 h, then pulverized into 300–600-μm particles. Finally, the SAP composite particles were washed 5 times in excess acetone to remove any unreacted monomers, oligomers, OS, and crosslinkers. In each reaction, the polymerization efficiency (yield) was at least 96%.

### 2.3. Preparation of SSAP

For the preparation of the SSAP, the surface-crosslinking solutions were prepared by mixing methanol (6 mL) and the selected surface crosslinker (0.5 mol.%). After mixing, distilled water (DW, 3 mL) was added to control the polarity of the solvent (2:1 methanol:water). The chemical structures and short-hand designations of the surface-crosslinkers are listed in Table 2. The CSAP composite (3 g) was allowed to swell for 10 min in the surface-crosslinking solution, then transferred to an aluminum tray, spread out thinly, and reacted at 140 °C for 15 min. Subsequently, the mixtures were washed with acetone for 10 min and dried at 60 °C in a convection oven for 30 min to remove any unreacted materials. The so-obtained surface-crosslinked CSAP composite is designated hereafter as the SSAP.

### 2.4. Fourier Transform Infrared (FT-IR) Characterization

The structures of the CSAP composites and the SSAP were characterized by Fourier transform infrared (FT-IR) spectroscopy using potassium bromide pellets on a Thermo Nicolet NEXUS 670 (NEXUS Instruments) over a scan range of 500–4000 cm^−1^ at a resolution of 1 cm^−1^.

### 2.5. Centrifuge Retention Capacity (CRC)

The centrifuge retention capacity (CRC) is a measure of the amount of fluid retained by the SAP after free absorption (i.e., swelling in the absence of pressure) followed by dehydration by centrifugation. Thus, a 100 mesh tea bag containing SAP (0.1 g) was immersed in excess saline solution for 30 min, then subjected to centrifugation at 300 G, and the water retention was measured. The CRC was then calculated from Equation (1):(1)$$\mathrm{CRC}=\frac{\omega_{1}-\omega_{0}}{\omega_{0}}$$
where ω_1_ and ω_0_ are the weights of the swollen and dry SAP, respectively.

### 2.6. Absorbency under Load (AUL)

The absorbency under load (AUL) is a measure of the amount of fluid absorbed by the SAP under a specific load and is presented in Figure 1. For this test, a ceramic filter and filter paper were sequentially placed on a petri dish, then a total of 0.9 g of the SAP was placed on the filter paper and fixed with a glass cylinder. A 0.3 psi Teflon pressure rod was then applied to the prepared device and sufficient saline solution was supplied to the petri dish. The AUL was then calculated using Equation (2):(2)$$\mathrm{AUL}=\frac{\omega_{1}-\omega_{0}}{\omega_{0}}$$
where ω_1_ and ω_0_ are the weights of the swollen and dry SAP, respectively.

### 2.7. Rheological Analysis of the Gel Strength

The gel strength was determined via rheological analysis using a Rotational rheometer (TA Instruments Ltd., ARES-G2) with a parallel plate geometry (25-mm plate diameter, 3-mm spacing). The SAP particles (0.2 g) were added to DW (10 mL) and allowed to swell for 30 min. The swollen gels were then placed on the parallel plate of the rheometer and the rheological properties were measured at 25 °C. The storage modulus (G’) and loss modulus (G”) were recorded at a constant shear strain of 0.2% over the frequency range of 0.1–100 Hz [21].

### 2.8. Permeability Measurement

Permeability is a measure of the amount of water that flows through the swollen SAP. First, SAP (0.5 g) was added to a stoppered chromatography column and allowed to swell for 30 min by the addition of saline solution (100 mL). A 0.3-psi pressure pillar was installed on the swollen SAP, the valve at the bottom of the column was opened, and the time required for 20 mL of saline solution to pass through was measured.

### 2.9. Re-Swellability Test

The re-swellability is a characteristic that can absorb water again after drying from the initial water absorption of SAP. The swollen SAP was dried in a convection oven at 70 °C for 24 h, then the CRC and AUL values were re-measured according to the above protocols. The swelling and re-drying steps were each performed four times (one experiment per day) to determine the cyclic re-swelling ability of the SAP particles.

## 3. Results

### 3.1. Preparation of the CSAP Composites

The CSAP composite was prepared as described in the Experimental section. Thus, after neutralization with NaOH, the radical polymerization of the vinyl monomers (IA and AA) in the presence of the diacrylate crosslinker (HDODA) results in the formation of a network structure. The chain propagation is initiated by thermal decomposition of ammonium persulfate to generate two sulfate radical anions via destabilization of the dioxygen bonds between the sulfate group [22,23]. The generated radicals are transferred to the vinyl group of the monomer to form a vinyl radical, which continues the chain propagation reaction. During chain propagation, the polymer chain reacts with the inner-crosslinker to form the crosslinked structure. The OS is added to the monomer mixture after the neutralization step and, hence, becomes embedded in the CSAP network via hydrogen bonding between the carboxyl or hydroxyl groups of the OS and the carboxyl groups of CSAP [24]. In addition, chemical bonding of the OS is possible via the formation of alkoxy radicals from the hydroxyl groups during chain propagation of the copolymer [25].

### 3.2. Preparation of the SSAP

The additional surface cross-linking of the CSAP composite is induced by reaction between those CSAP carboxyl groups that did not participate in the neutralization reaction and the epoxide groups of the surface-crosslinker. This reaction results in the formation of a repeating unit that contains both ester and a hydroxyl functional groups [26]. The surface-crosslinking of the CSAP composite was performed using four different bifunctional surface-crosslinkers with epoxide groups at both ends, namely: BADGE, PEGDGE, EGDGE, and DGE (Table 2). A schematic diagram of the surface-crosslinking process with each crosslinker is presented in Figure 2.

### 3.3. Characterization of the CSAP Composite and the SSAP

The FT-IR spectra of the IA, AA, OS, CSAP composite, and SSAP are presented in Figure 3. Here, the broad absorption band at 2900–3350 cm^−1^ is attributed to the OH stretching vibrations of the carboxyl groups of IA, AA, and the hydroxyl groups of OS. The absorption peak at 2853 cm^−1^ reveals the presence of CH_2_ in the polymer backbone, while that at 1709 cm^−1^ is due to the carboxyl group C=O stretching vibrations in IA, AA, and OS. Meanwhile, the peaks at 911 cm^−1^ and 1638 cm^−1^ represent the C=CH_2_ vinyl groups of IA and AA, respectively. In addition, the CSAP and SSAP exhibit peaks at 1584 cm^−1^ and 1386 cm^−1^ due to the carboxylate (COO^−^) ion generated by neutralization of the carboxyl groups. Since carboxylate has a resonance structure, the C=O stretching vibrations are divided into symmetrical and asymmetrical peaks [27]. The band at 1690 cm^−1^ in the spectrum of OS is assigned to the stretching vibrations of the carbonyl groups due to oxidation of the starch molecule. An absorption peak is also observed at 1015 cm^−1^, which is attributable to the C–O–C stretching vibrations of starch. These results demonstrate the successful synthesis of SAP from IA and AA, and the incorporation of OS into the matrix.

### 3.4. The Effects of OS Content upon the CRC, AUL, and Gel Strength

The changes in the CRC and AUL values of the CSAP composites as a function of OS content are presented in Figure 4a. Here, the CRC value (the green line with square symbols) is seen to increase steadily with increasing OS content, up to a maximum of 60.2 g/g at 4.0 g OS, and to decrease thereafter. The initial increase is attributed to hydrogen bonding between the adsorbed water and the hydrophilic hydroxyl groups in OS, thus limiting the amount of water that could be removed by centrifugal force. However, the subsequent decrease in the CRC at higher OS content occurs because the introduction of excess hydroxyl groups results in aggregation of the OS units in preference to the formation of hydrogen bonds with water, thus adversely affecting the expansion of the SAP network and decreasing the absorption capacity. A similar trend is observed for the AUL values of the CSAP composites with increasing OS content (the blue line with triangle symbols in Figure 4a).

The effects of varying OS content on the gel strengths of the CSAP composites are indicated in Figure 4b. Here, the gel strength is seen to increase with increasing OS content. As with the increases in the CRC and AUL, this is attributed to an increase in hydrogen bonding with increased input of OS [28]. However, the CSAP composites with excess OS (e.g., the IcA_OS_4.5_ and IcA_OS_5.0_) exhibit a decrease in the AUL due to aggregation of the OS caused by excess hydroxyl groups, along with a decrease in absorption capacity as the internal network becomes filled with excess OS. In addition, the standard deviation from three separate measurements on different OS content of CSAPs was around 5% in all samples. The change in standard deviation are prominent in the aging process based on the structural rearrangement hypothesis of the filler [29]. However, the same result within the error range showed that the sample was successfully measured under the same storage conditions.

In view of the high CRC and AUL values of the IcA_OS_4.0_ (60.2 g/g and 11.1 g/g, respectively), this was selected as the optimal CSAP composite, and was used for the preparation of SSAPs via subsequent surface-crosslinking reaction.

### 3.5. The Effect of the Type and Content of Surface-Crosslinker

The changes in the CRC and AUL values of the CSAP composite and the SSAP according to the choice of surface-crosslinker are presented in Figure 5. It is thought that the surface crosslinking reaction using BADGE did not completely proceed. In contrast to the other types of surface-crosslinker, BADGE contains cyclic aromatic groups that may sterically prevent it from penetrating into the CSAP composite, so that the reaction proceeds only on the outermost surface of the CSAP. Hence, little change is observed in the CRC and AUL values compared to those of the CSAP composite. The greatest overall changes in both the CRC and AUL values of the SSAP relative to those of the CSAP are obtained using the crosslinker with the lowest molecular weight, i.e., the DGE. Indeed, this crosslinker provides the SSAP with the highest AUL value of all, thus indicating the highest gel strength. However, the highest CRC value is obtained using the crosslinking agent with the highest molecular weight, i.e., the PEGDGE. This contrasting trend may be attributed to the different surface crosslinking densities of the obtained SSAPs, as the distance between crosslinking points is dictated by the difference in chain lengths (and, hence, the molecular weights) of the surface crosslinkers. Moreover, since commercial SAPs for use in disposable diapers require CRC and AUL values of 30 g/g and 20 g/g, respectively [30], the present results suggest that EGDGE is the best surface-crosslinker for the fabrication of CSAP composites from OS.

The changes in the CRC and AUL values of the SSAP of EGDGE content are presented in Figure 6. Here, the CRC value is seen to decrease with increasing EGDGE content. On the other hand, AUL value increases. According to the Flory-Rehner correlation, the crosslinking density is directly related to gel strength [31]. Thus, an increase in the content of surface-crosslinker leads to an increase in the crosslinking density and gel strength of the SSAP, but is accompanied by a decrease in the CRC. These results demonstrate the importance of controlling the surface-crosslinking density of SSAP in order to achieve the optimum performance.

The changes in the permeability of the CSAP composite and the SSAPs according to the type of surface crosslinker and the content of EGDGE are presented in Figure 7. For the same content of surface crosslinker, the SSAP with the fastest permeability is obtained using the crosslinker with the lowest molecular weight (i.e., the DGE). However, a slight increase in the permeability of the SSAP is observed with increasing content of EGDEG. These results indicate that an increase in the surface-crosslinking density leads to an increase in the amount of fluid flowing between the SAP particles and alleviates the gel-blocking phenomenon.

### 3.6. The Rheological Analysis of Gel Strength

The storage modulus (G’) and loss modulus (G”) versus frequency for the CSAP composite (IcA_OS_4.0_) and the SSAP obtained using 0.75 mol EGDGE are presented in Figure 8. Here, CSAP composite and SSAP each showed a dynamic modulus independent of the applied frequency with G’ higher than G”. It demonstrates that the SAPs are pure elastomer [32]. Moreover, the SSAP exhibits a significantly larger G’ than that of the CSAP composite, thus demonstrating that surface-crosslinking enhances the gel strength of the CSAP composite. Furthermore, the standard deviation decreased from 5% of CSAP to 3% of SSAP. This means that the bounding between OS and the polymer network was strengthened due to the additional crosslinking.

### 3.7. Re-Swellability

The results of the re-swellability analysis for the CSAP composite (IcA_OS_4.0_) and the SSAP that was obtained using 0.75 mol EGDGE are presented in Figure 9. Here, the re-swelling ratios in CSAP indicate a 73% decrease in the CRC, and a 51% decrease in the AUL, at the second water intake (Figure 9a). Subsequently, the CSAP was dissolved in aqueous saline solution and a gel was not observed at the third time of water intake. By contrast, the SSAP exhibits decreases of approximately 10% and 13% in the CRC and AUL values, respectively, at the first swelling test, compared to those obtained at the initial water intake (Figure 9b). This may be attributed to the good elastic modulus of the expanded SSAP polymer chains due to the high gel strength provided by the surface crosslinking reaction. Hence, the SSAP shows the potential for application in the re-recycling of diaper products in agriculture.

## 4. Conclusions

Itaconic acid (IA)-based core-SAP (CSAP) composites were prepared by the incorporation of various proportions of oxidized starch (OS). The successful syntheses of the OS-containing CSAP composites were demonstrated by FT-IR spectroscopy and rheological analysis. The polymer yields were found to be above 96%. A surface-crosslinked SAP (SSAP) with enhanced gel strength was then manufactured via surface-crosslinking of the as-prepared CSAP, and the effects of the epoxy-based crosslinking agent upon the degree of crosslinking and absorption properties were investigated. The examined surface crosslinkers included bisphenol a diglycidyl ether (BADFE), poly(ethylene glycol) diglycidyl ether (PEGDGE), ethylene glycol diglycidyl ether (EGDGE), and diglycidyl ether (DGE).

The change in absorption characteristics due to surface-crosslinking is closely related to the molecular weight of the surface-crosslinker, such that the increase in absorbency under load (AUL) compared to that of the CSAP was highest with DGE. However, this resulted in the greatest decrease in the centrifuge retention capacity (CRC). In view of the changes in CRC and AUL, the optimal surface-crosslinker was found to be EGDGE. For potential application in a commercial SAP, the EDGE content of the SSAP was varied and the optimum performance was obtained with 0.75 mol EGDGE. The successful surface crosslinking reaction was confirmed via a comparison of the gel strengths of the CSAP and SSAP by rheological analysis. In addition, a comparison of the permeabilities confirmed that the gel-blocking phenomenon was alleviated. Finally, repeated swelling and drying of the CSAP and SSAP confirmed that the SSAP could be subjected to four cycles of re-swelling while retaining a high gel strength. By contrast, the CSAP exhibited a low gel strength and a dramatic decrease in absorption properties, such that the gel form could not be maintained at the third re-swelling. Hence, it is believed that the SSAP with improved gel strength is suitable for commercialization in disposable diapers.

## Figures and Tables

**Figure 1 polymers-13-02859-f001:**
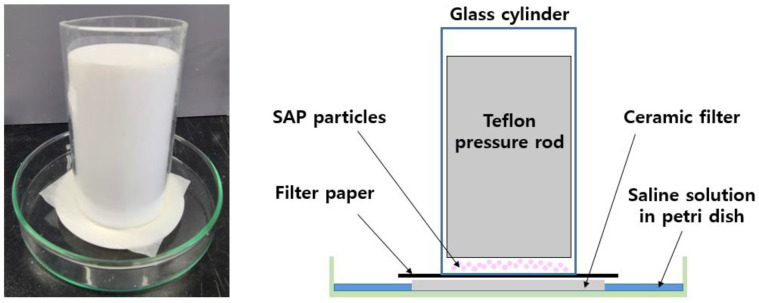
AUL equipment and schematic diagram.

**Figure 2 polymers-13-02859-f002:**
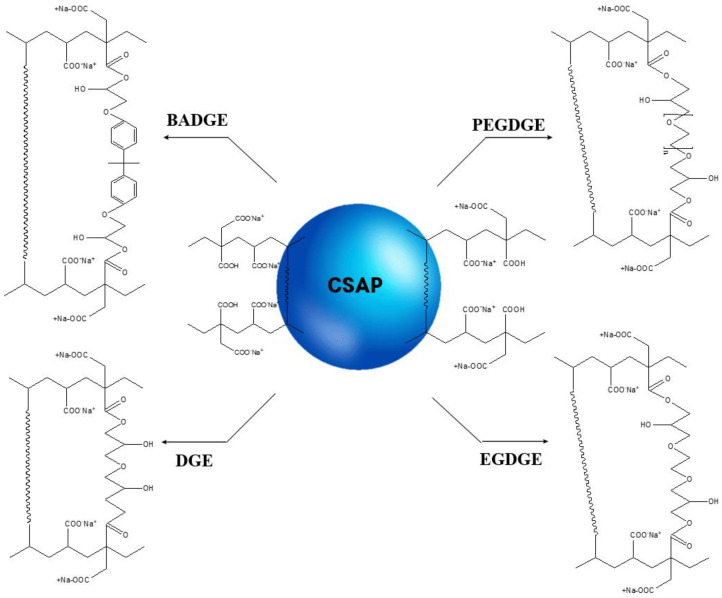
A schematic diagram of the surface-crosslinking reaction with various crosslinkers.

**Figure 3 polymers-13-02859-f003:**
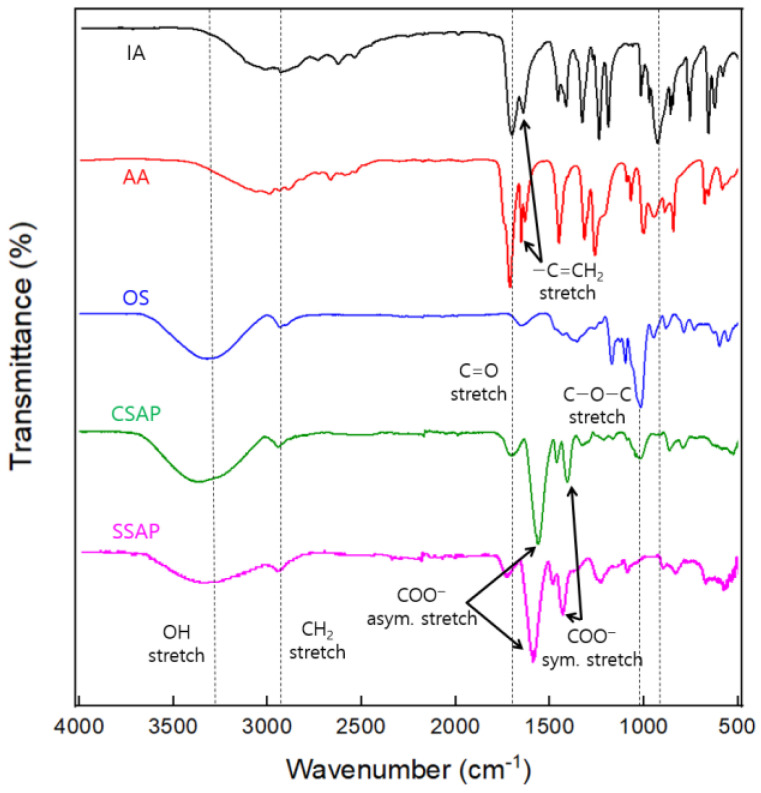
The FT-IR spectra of the IA, AA, OS, CSAP composite, and SSAP.

**Figure 4 polymers-13-02859-f004:**
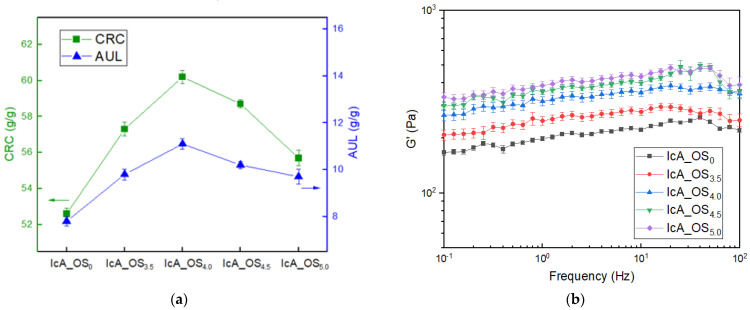
The effects of OS content on (**a**) the absorption properties (CRC and AUL), and (**b**) the gel strength of the CSAP composites. The error bars represent standard deviations of the mean from three separate measurements.

**Figure 5 polymers-13-02859-f005:**
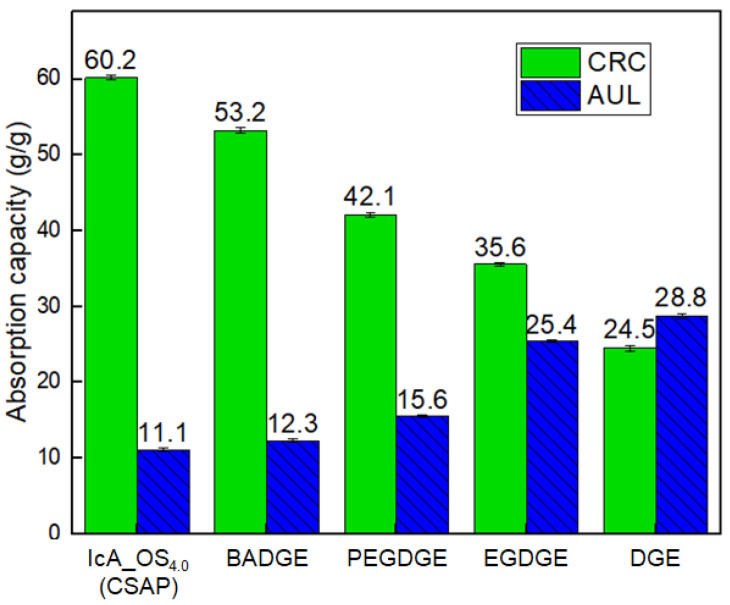
The changes in the CRC and AUL values of the CSAP composites and SSAP according to the choice of surface-crosslinker.

**Figure 6 polymers-13-02859-f006:**
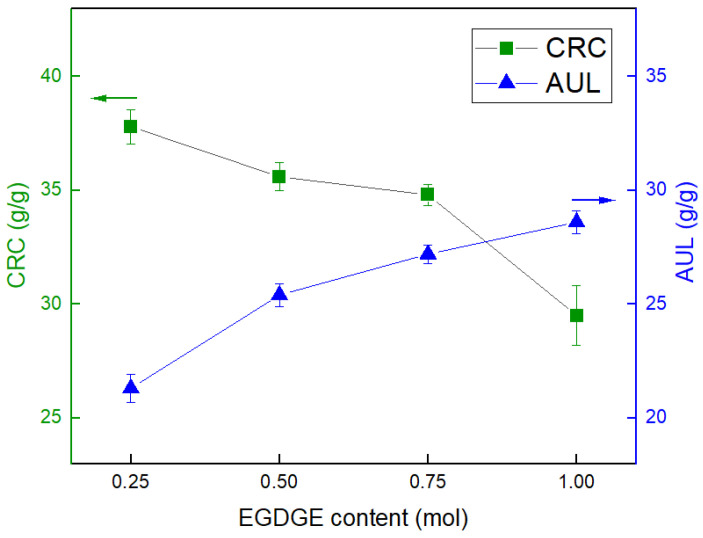
The effects of EGDGE content on the CRC and AUL values of the SSAP.

**Figure 7 polymers-13-02859-f007:**
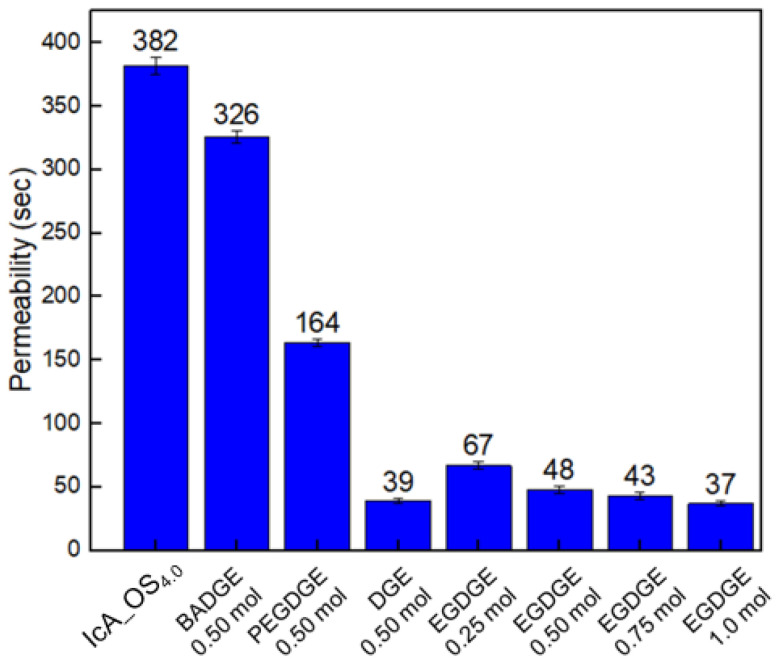
The changes in the permeability of the CSAP composite (IcA_OS_4.0_) and the SSAP according to the choice of surface-crosslinker and the content of EGDGE.

**Figure 8 polymers-13-02859-f008:**
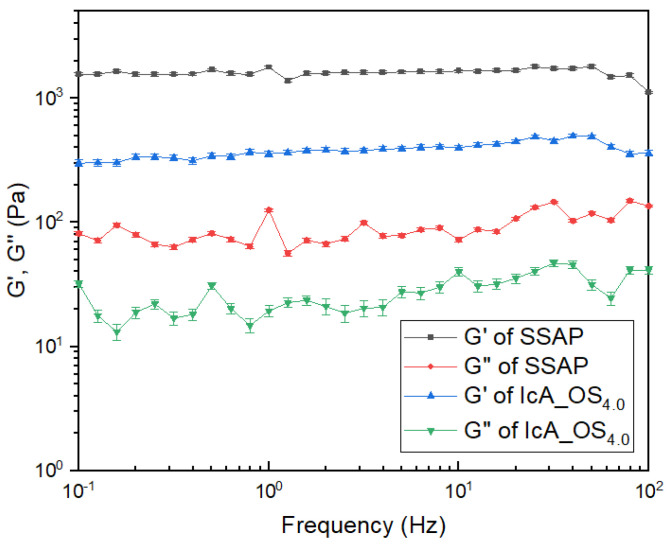
The G’ and G” versus frequency for the CSAP composite (IcA_OS_4.0_) and the SSAP obtained using 0.75 mol EGDGE. The error bars represent standard deviations of the mean from three separate measurements.

**Figure 9 polymers-13-02859-f009:**
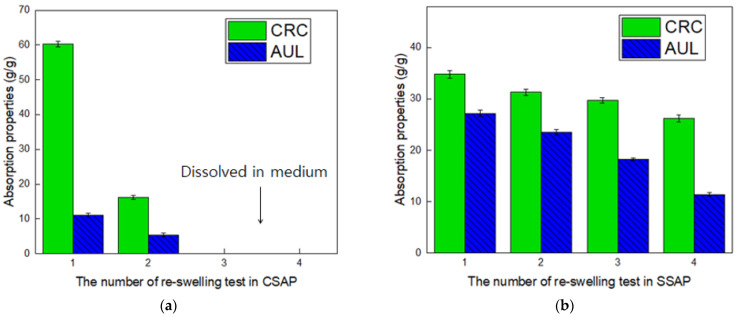
The absorption properties according to the number of re-swelling repetitions for (**a**) the CSAP composite (IcA_OS_4.0_) and (**b**) the SSAP obtained using 0.75 mol EGDGE.

**Table 1 polymers-13-02859-t001:** Synthetic composition of CSAP composites.

Sample	IA (mol)	AA (mol)	OS (g)	Neutralization Degree (%)	HDODA (mmol%)	APS (wt.%)
IcA_OS_0_	0.5	0.5	0	75	2.0	0.5
IcA_OS_3.5_			3.5			
IcA_OS_4.0_			4.0			
IcA_OS_4.5_			4.5			
IcA_OS_5.0_			5.0			

**Table 2 polymers-13-02859-t002:** Surface-crosslinking solution conditions.

Methanol(mL)	DW(mL)	Surface-Crosslinker
Designation	Chemical Structure	Content(mol%)
6	3	BADGE	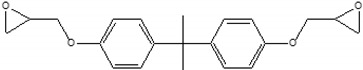	0.5
PEGDGE	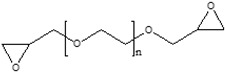	0.5
EGDGE	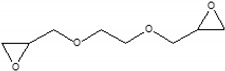	0.5
DGE	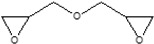	0.5

## Data Availability

Data sharing not applicable.

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
