# Peer review of "Enhancement of Gel Strength of Itaconic Acid-Based Superabsorbent Polymer Composites Using Oxidized Starch"

_polymers, 2021, doi:10.3390/polym13172859_

Round 1
Reviewer 1 Report
The work entitled “Enhancement of gel strength of itaconic acid-based superabsorbent polymer composites using oxidized starch” by Kim et al. introduces a new to develop a superabsorbent polymer composites using oxidized starch. The introduction of the subject and the methodology are very simple to understand and follow. The authors did a good job highlighting the main features of their strategy, implementing it, and describing the steps to achieve it. I would recommend the authors to give a little more emphasis on the novelty in the introduction section. That would be important to elevate the status of the manuscript as a first report of its kind. The data collected is also of easy comprehension and the discussion is quite simple. In that particular case I would recommend the authors to deepen their findings by criticizing them with the literature as it is not done properly. The authors kept the analysis a bit superficial, and it can be further highlighted if compared to less or as successful strategies similar to this and how is improved above those. This would be very important for the readers. I would also recommend the addition of standard deviations in Fig 3/7 and statistical analysis in Fig 4/8. The conclusions are very to the point and concise. Overall, the manuscript has scientific merit (the authors did a great job) and is very up-to-date, and, thus, should be considered for publication after these small details are fixed.
Author Response
Response to Reviewer 1 Comments
The work entitled “Enhancement of gel strength of itaconic acid-based superabsorbent polymer composites using oxidized starch” by Kim et al. introduces a new to develop a superabsorbent polymer composites using oxidized starch. The introduction of the subject and the methodology are very simple to understand and follow. The authors did a good job highlighting the main features of their strategy, implementing it, and describing the steps to achieve it.
Point 1: I would recommend the authors to give a little more emphasis on the novelty in the introduction section. That would be important to elevate the status of the manuscript as a first report of its kind.
Response 1: As you requested, we revised the introduction section to emphasize the novelty.
Point 2: The data collected is also of easy comprehension and the discussion is quite simple. In that particular case I would recommend the authors to deepen their findings by criticizing them with the literature as it is not done properly. The authors kept the analysis a bit superficial, and it can be further highlighted if compared to less or as successful strategies similar to this and how is improved above those. This would be very important for the readers.
Response 2: We strengthened the scientific analysis in revised manuscript.
Point 1: I would also recommend the addition of standard deviations in Fig 3/7 and statistical analysis in Fig 4/8. The conclusions are very to the point and concise. Overall, the manuscript has scientific merit (the authors did a great job) and is very up-to-date, and, thus, should be considered for publication after these small details are fixed.
Response 3: We inserted an error bar in Figure 3/7. These error bars represent the standard deviation of the mean through three separate measurements. We also represented the statistical analysis of Figure 4/8 using error bars which were an average through 3 repetitions of the same sample.
Overall, the manuscript has scientific merit (the authors did a great job) and is very up-to-date, and, thus, should be considered for publication after these small details are fixed.
Reviewer 2 Report
This manuscript does not correspond to the high-quality criteria of the journal of Polymers. Therefore, I recommend rejecting this manuscript. The novelty level of this manuscript is very low. Although it is adequately written, it offers no critical information and no new slant on the review topic. Most of the content in the manuscript is better for the journal with a lower level. The introduction section should be improved. Materials characterization tests are not enough. Interpretation of tests is very poor and requires re-analysis and further testing.
Author Response
Response to Reviewer 2 Comments
This manuscript does not correspond to the high-quality criteria of the journal of Polymers. Therefore, I recommend rejecting this manuscript. The novelty level of this manuscript is very low. Although it is adequately written, it offers no critical information and no new slant on the review topic. Most of the content in the manuscript is better for the journal with a lower level.
Point 1: The introduction section should be improved. Materials characterization tests are not enough. Interpretation of tests is very poor and requires re-analysis and further testing.
Response 1:
To emphasize novelty, we revised the introduction section. We added schematic of AUL test method. Furthermore, we also added the standard deviation in the rheological analysis for measuring the gel strength. Thank you for your helpful advice.

Reviewer 3 Report
In this work, the core-superabsorbent polymer hydrogels (CSAP) were prepared by the random copolymerization of IA and AA with oxidized starch (OS) as the filler. To enhance the gel strength, SSAP gel particles with core-shell SAP structures were prepared via the further crosslinking process on the surface of CSAP. The effect of OS content, crosslinker density, and type of crosslinkers on the absorption properties and rheological behavior were systematically investigated in this manuscript. By optimized the composition, SSAP gel particles with the good absorption properties and strength were obtained. There are number of problems to be solved before publication, including presentation, interpretation, scientific logical discussions. Specific comments are as follows:
- Page 4, line 110. The experimental condition is not clear. For example, it is difficult to understand how to measure the absorbency under load (AUL). It is better to draw the diagrammatic sketch to illustrate the method of AUL. What is the concentration of saline solution for the AUL and CRC measurement?
- Page 9, line 251. The author stated that “the higher G’ than G” in each case indicates that the viscous behaviors of the CSAP composite…”. While, the dynamic modulus independent of the applied frequency in Figure 7 indeed demonstrates that the gels are pure elastic.
- It is difficult to understand the trade-off behavior between the CRC and AUL in Figure 4 and 5.
- Although SSAP could be subjected to re-swelling behavior, the water uptake decreased with the numbers of the cyclic reswelling. What is the mechanism behind this phenomenon?
Author Response
Response to Reviewer 3 Comments
In this work, the core-superabsorbent polymer hydrogels (CSAP) were prepared by the random copolymerization of IA and AA with oxidized starch (OS) as the filler. To enhance the gel strength, SSAP gel particles with core-shell SAP structures were prepared via the further crosslinking process on the surface of CSAP. The effect of OS content, crosslinker density, and type of crosslinkers on the absorption properties and rheological behavior were systematically investigated in this manuscript. By optimized the composition, SSAP gel particles with the good absorption properties and strength were obtained. There are number of problems to be solved before publication, including presentation, interpretation, scientific logical discussions. Specific comments are as follows:
Point 1: Page 4, line 110. The experimental condition is not clear. For example, it is difficult to understand how to measure the absorbency under load (AUL). It is better to draw the diagrammatic sketch to illustrate the method of AUL. What is the concentration of saline solution for the AUL and CRC measurement?
Response 1: We inserted the diagrammatic sketch of AUL. And the concentration of saline solution for the AUL and CRC is 0.908% (0.9 wt% NaCl solution).
Point 2: Page 9, line 251. The author stated that “the higher G’ than G” in each case indicates that the viscous behaviors of the CSAP composite…”. While, the dynamic modulus independent of the applied frequency in Figure 7 indeed demonstrates that the gels are pure elastic.
Response 2: You're right. Thank you for your advice. We made a mistake and corrected.
Point 3: It is difficult to understand the trade-off behavior between the CRC and AUL in Figure 4 and 5.
Response 3: The trade-off behaviour between the CRC and AUL is generally accepted in SAP industry. However, it was not the case in Figure 4 and 5, we deleted that sentence.
Point 4: Although SSAP could be subjected to re-swelling behavior, the water uptake decreased with the numbers of the cyclic reswelling. What is the mechanism behind this phenomenon?
Response 4: SAP is not a perfectly elastic material, and loss modulus exists as shown in Figure 7 (Figure 8 in the revised manuscript). This means that the maximum swelling cannot be achieved with repeated re-swelling. Also, although not mentioned in this paper, we think it is related to osmotic pressure i.e., absorption and dehydration of water may cause the initial osmotic pressure to be out of balance.

Round 2
Reviewer 1 Report
The authors implemented the necessary changes and the manuscript is now ready to publish.
Reviewer 2 Report
This manuscript does not correspond to the high-quality criteria of the journal of Polymers.
Reviewer 3 Report
The present version is suitable for the publication.